# Green Production of Functionalized Few-Layer Graphene–Silver Nanocomposites Using Gallnut Extract for Antibacterial Application

**DOI:** 10.3390/mi13081232

**Published:** 2022-07-31

**Authors:** Yingjie Bu, Anamika Kushwaha, Lalit Goswami, Beom-Soo Kim

**Affiliations:** Department of Chemical Engineering, Chungbuk National University, Cheongju 28644, Chungbuk, Korea; byj17@chungbuk.ac.kr (Y.B.); kushwaha.anamika@gmail.com (A.K.); lalitgoswami660323@gmail.com (L.G.)

**Keywords:** gallnut extract, high-shear exfoliation, Raman spectra, antibacterial activity

## Abstract

Recently, there has been much attention paid to functionalized few-layer graphene (FFG) owing to its many biomedical applications, such as in bioimaging, biosensors, drug delivery, tissue scaffolds, nanocarriers, etc. Hence, the preparation of FFG has now become of great interest to researchers. The present study systematically investigates the utilization of gallnut extract (GNE) during the process of high-shear exfoliation for the efficient conversion of expanded graphite to FFG. Various parameters, such as GNE concentration, graphite concentration, exfoliation time, and the rotation speed of the high-shear mixer, were initially optimized for FFG production. The prepared FFG was characterized in terms of surface functionality and morphology using Raman spectra, X-ray diffraction (XRD), X-ray photoelectron spectroscopy (XPS), transmission electron microscopy, and scanning electron microscopy analyses. Further, the conjugation of FFG with Ag was confirmed by XRD, XPS, and energy-dispersive X-ray spectra. The Ag–FFG composite exhibited antibacterial activity against both Gram-positive and Gram-negative bacteria through the agar well diffusion method. This study provides an efficient, economical, and eco-friendly FFG and Ag–FFG production method for biomedical applications.

## 1. Introduction

Graphene is an ultrathin, typically hydrophobic, chemically inert, and well-recognized two-dimensional (2D) honeycomb sp^2^ carbon lattice [1,2,3]. It is an immensely studied nanomaterial owing to its unprecedented chemical and physical properties [4,5]. Since its first discovery in 2004 by Geim and Novoselov utilizing the ‘Scotch Tape’ procedure, graphene has been frequently isolated, explored, and characterized for its unique electronic properties. Isolated graphene possesses exclusive properties such as a high charge carrier mobility of 2000–5000 cm^2^/V s [6]. Nowadays, graphene is applied in high-performance and multifunctional polymer-based nanocomposites, possessing an exceptional mechanical, optical, electrical, and thermal conductivities, and chemical stability, as well as a large surface area [6,7,8,9]. It has numerous applications in energy storage, sensing, flexible electrodes, smart coatings, biomedical, bioimaging, etc. [9,10,11]. Thus, to allow graphene-based nanocomposites to reach their potential, a facile and scalable production of high-quality graphene at an economic value is still a prerequisite [8,10].

Presently, graphene is produced via two ways: (i) bottom-up and (ii) top-down methods [12]. The bottom-up approach includes chemical vapor deposition, which produces defect-free graphene along with a homogenous thickness of the layer and enables high yields of graphene monolayers [13,14]. The only limitation of this method is the high cost of production. In comparison, the top-down approach includes wet chemical synthesis or mechanical exfoliation from graphite, which are better options for producing low-cost graphene [15]. Further, different covalent and non-covalent approaches have been reported for the conversion of graphene oxide (GO) into functionalized graphene using various chemicals and strong acids and alkalis, molten salts, and electrochemical intercalations (sodium borohydride, hydrazine, hydroquinone, etc.) [16]. Initially, chemicals are added to the graphite solution, followed by sonication or exfoliation. Further, several surface functionalization techniques for graphene from GO have been utilized using polystyrene sulfonate, aryl diazonium salt, conjugated polyelectrolyte, p-phenylene diamine, etc. [8]. Most of these chemicals are toxic, cost-intensive, lethal to plants and animals, and have low exfoliation yields, which are the main limitations of their application [17].

To overcome these shortcomings, researchers are trying to develop novel and green reduction processes to synthesize reduced/functionalized GO from GO by utilizing numerous bio-based molecules and materials such as ascorbic acid, citric acid, tannic acid, curcumin, reducing sugars, gum arabic, artemisinin, gallic acid, etc. [17,18]. However, limited studies have reported on the utilization of plant extracts for converting graphite to graphene [19,20]. In addition, some modifications/improvisations were examined to enhance the electrical, thermal, and catalytic properties of graphene by impregnating essential metals onto the surface [21]. Several processes have been developed to allow the in situ attachment of metal nanoparticles onto the surface of functionalized graphene, which has many important applications [21,22,23].

In addition, the emergence of nanotechnology has led to the development of nanomaterials including nanoparticles possessing antimicrobial characteristics, such as silver nanoparticles (AgNPs). Such nanoparticles could be seen as an alternative option for mitigating infectious diseases and a promising solution for enhancing antibiotic resistivity. Numerous physical, chemical, and biological procedures have already been reported for the synthesis of AgNPs [24,25,26,27,28]. Out of the relevant chemical processes, the reduction of Ag salt solutions using reducing agents is the simplest and most economical, and is often applied. However, the reduction of Ag+ to Ag° leads to their agglomeration into oligomeric clusters and further leads to the formation of colloidal AgNPs. Thus, several stabilizing agents are added for AgNPs synthesis to control the shapes and sizes of nanoparticles. Nowadays, functionalized few-layer graphene (FFG)-based nanocomposites have emerged as suitable functional materials to disperse and stabilize Ag owing to their huge specific surface areas and ample oxygenated functional groups. In addition, FFG can be easily dispersed and is stable in water. These oxygen-containing functional groups act as anchoring sites for the attachment of AgNPs; hence, they are a perfect support for the growth and stabilization of AgNPs [24,27].

Henceforth, the present investigation focuses on the optimization and production of FFG using the high-shear exfoliation process. Here, exfoliation was carried out using gallnut extract (GNE). Various parameters such as GNE concentration, graphite concentration, exfoliation time, and the rotation speed of the high-shear mixer were initially optimized for FFG production. The prepared FFG was characterized for surface functionality and morphology using Raman spectra, X-ray diffraction (XRD), transmission electron microscopy (TEM), and scanning electron microscopy (SEM) analyses. In addition, the conjugation of FFG with Ag was confirmed by XRD, X-ray photoelectron spectroscopy (XPS), and energy-dispersive X-ray (EDX) spectra. Further, the in situ growth of AgNPs onto the FFG surface was carried out without utilizing any reducing agent and was evaluated for antibacterial application.

## 2. Materials and Methods

### 2.1. Materials

Expanded graphite was purchased from Samjung C & G, Korea. Silver nitrate (CAS no. 7761-88-8 ≥ 99.8% purity) and gallnut powder were procured from Samchun Pure Chemical Co., Ltd., Seoul, Korea and Hamyoung Co., Seoul, Korea, respectively.

### 2.2. Synthesis of FFG

#### 2.2.1. Preparation of Gallnut Extract 

Initially, gallnut extract was prepared using 1 g of gallnut powder that was placed into an electric brewing pot containing 100 mL of distilled water and it was boiled at 100 °C for 90 min [17]. Then, it was centrifuged at 13,000 rpm for 20 min to remove suspended particulate matter. The supernatant was stored at 4 °C until further use.

#### 2.2.2. Exfoliation of FFG Using Gallnut Extract

FFG was prepared from expanded graphite using the L5M high-shear laboratory mixer exfoliation process, as described by Paton et al. [29], with some minor modification. In brief, graphite was added into the mixing vessel (500 mL) along with gallnut extract (in different concentrations) followed by gradual increasing the mixing speed. The mixer was allowed to run for a predetermined mixing time. To avoid the high-shear exfoliation system heating up, the mixing vessel was kept in a water bath at 10 °C. A typical mixer with a 32 mm rotor and an axial flow head (100 mm diameter) was utilized in a vertically upward orientation to reduce aeration and maintain heavy insoluble suspended solids in the system [29]. Figure 1 represents a brief schematic of the FFG preparation process. To evaluate the effect of various parameters on the exfoliation process, four different parameters, i.e., GNE concentration (2–10 g/L), graphite concentration (5–30 g/L), exfoliation time (2–8 h), and rotational speed (3000–6000 rpm), of the high-shear mixer were examined. For the optimization of rotational speed and exfoliation time, the exfoliation process was further qualitatively examined via Raman spectroscopy, while the effects of GNE concentration and initial graphite concentration were analyzed through the obtained graphene concentration.

After the exfoliation step, the solution was centrifuged at 3,000 rpm for 30 min to separate large flakes. The supernatant from this solution was collected and centrifuged to obtain FFG. To remove unbound extracts or free phytochemicals, the FFG nanosheets were repeatedly washed with distilled water, centrifuged at 13,000 rpm for 20 min, and finally the precipitated sheets were freeze-dried to obtain FFG powder.

### 2.3. Preparation of Ag–FFG Nanocomposites and Their Antibacterial Activity

To prepare Ag–FFG nanocomposites, 50 mL of GNE functionalized FFG dispersion solution (1 g/L) was stirred at 200 rpm (60 °C), and then 25 mL of AgNO_3_ (1 g/L) was added. The lower the temperature, the smaller the concentration of Ag anchored on the FFG surface [26]. The mixture was continuously stirred for 24 h to completely reduce Ag^+^ ions onto the prepared FFG. The addition of GNE acted as both an exfoliating agent and in situ functionalization site, enabling the reduction of Ag^+^ ions. After this, the dispersion was centrifuged at 13,000 rpm for 20 min and the precipitated pellet was washed four times to remove the unreacted AgNO_3_. Ag–FFG nanocomposites were freeze-dried and further used for characterization and antibacterial experiments.

The antibacterial activity of the prepared Ag–FFG nanocomposites was studied using the agar well disk diffusion method against different bacterial strains (belonging to four genera: *Staphylococcus aureus* ATCC 6538 (Gram-positive), *Escherichia coli* ATCC 11775 (Gram-negative), *Bacillus megaterium* KCTC 2194 (Gram-positive), and *Pseudomonas aeruginosa* PR3 (NRRL strain B-18602) (Gram-negative). All bacteria were initially cultured on nutrient agar medium and incubated at 37 °C for 24 h. Bacterial culture (100 μL) of optical density (OD) 1.0 was spread on Luria Bertani (LB) agar plate. In each Petri plate, three wells with 110 mm diameters were corkscrewed and filled with 100 µL each of FFG, Ag salt, and Ag–FFG. Plates were kept inside the incubator at 37 °C for 24 h. After incubation, the inhibition zones on the plates were recorded as inhibition against microbial species [30,31,32].

### 2.4. Characterization of FFG and Ag–FFG Nanocomposites 

Various techniques were used to determine the quality of FFG obtained from expanded graphite using a high-shear exfoliation approach.

#### 2.4.1. Raman Spectroscopy and X-ray Diffraction Analysis

Raman spectroscopy was used to determine the qualitative conversion of graphite to FFG. Here, the exfoliation time and rotation speed were optimized for FFG preparation. Two characteristic peaks of graphene with different peak shapes and intensities, i.e., G (1350 cm^−1^) and 2D bands (2700 cm^−1^), were examined [33]. A Raman 2D spectrum was also used to measure the flake thickness of the produced FFG suspension [29,34]. In addition, the crystallinity structures of graphite and FFG were studied using the Raman spectrum. The ratio of the intensity (M) of the 2D band for the prepared FFG was evaluated at the wavelength connecting with the peak of the initial graphite 2D band $\left(\omega_{peak,\;Graphite}\right)$, which can be easily identified at 2729 cm^−1^, divided by the intensity at the wavelength connecting with the low-energy shoulder of the initial graphite 2D band defined as $\omega_{shoulder,Graphite}=\omega_{peak,\;Graphite\;}-30\;{\mathrm{cm}}^{-1}$ [29]. To normalize the metric parameter, the ratio of intensities for FFG was divided by that of the initial graphite. Here, the Raman spectra of all samples were scanned in the range of 100–3500 cm^−1^ using a LabRam HR, Horiba, Japan.
(1)$$M=\frac{I_{FFG\left(\omega_{peak,Graphite}\right)}/I_{FFG\left(\omega_{shoulder,Graphite}\right)}}{I_{Graphite\left(\omega_{peak,Graphite}\right)}/I_{Graphite\left(\omega_{shoulder,Graphite}\right)}}$$

Once the metric parameter *M* is known, the number of monolayers per FFG flake, *N_G_,* can be determined [29,35]:(2)$$N_{G}=10^{0.84\times M+0.45\times M^{2}}$$

Further, graphene sheets that are totally without plane defects show an intensity ratio for the *D* and *G* bands, which is closely related to the mean lateral flake size (*L*) by:(3)$$\frac{I_{D}}{I_{G}}\approx {\left(I_{D}/I_{G}\right)}_{Graphite}+\left(\frac{k}{L}\right)$$
where *(I_D_/I_G_*)_Graphite_ is the *D* to *G* band intensity ratio connected with the initial graphite and *k* is a parameter reported as ∼0.17 [36,37]. This can be used to estimate the FFG nanosheet size from the Raman spectrum.

The XRD profile was recorded using JP/SmartLab at 45 kV and 40 mA in the 2θ range of 7–100° at a scale rate of 0.05° per 0.5 s. The interlayer distance between graphene sheets was determined using Bragg’s law, Equation (4) [38]:(4)nλ=2dsinθ 
where *n* = 1, *θ* is the Bragg’s angle, *d* is the spacing of the diffracting planes, and *λ* is the X-ray wavelength used (Cu/Kα = 0.154 nm).

In addition, based on Scherrer’s equation, Equation (5), the crystal sizes of FFG and Ag–FFG were estimated [17]:(5)$$\mathrm{Crystal}\;\mathrm{size}\;\left(\mathrm{nm}\right)=\frac{k\times \lambda }{d\times cos\theta }\;$$
where *k* is the shape factor, *θ* is the half value of angle of corresponding peak, and *d* is the full width at half the maximum intensity.

XPS was used to analyze the surface chemistry of Ag–FFG using PHI Quantera-II and individual component spectra were deconvoluted using XPS PEAK41 software [39].

#### 2.4.2. Microscopy Analysis

The surface morphology and structure of the prepared FFG were characterized at ultrahigh resolution using TEM and SEM. For TEM analysis, LEO-1530 was utilized (Libra 120, Carl Zeiss, Oberkochen, Germany) at 120 kV. Samples were initially drop cast onto carbon-coated copper grids and air-dried overnight prior to analysis. For SEM analysis (Leo-1530, Carl Zeiss, Oberkochen, Germany), samples (FFG and Ag–FFG) were mounted on aluminum stubs using double-stick carbon tape with a fine double layer platinum coating prior to analysis [40]. Further, to confirm the presence of Ag on the surface of FFG, EDX analysis was performed on both samples.

## 3. Results and Discussion

### 3.1. Preparation and Characterization of FFG

For the conversion of expanded graphite to FFG, a high-shear exfoliation method was utilized. Initially, gallnut extract containing tannic acid was supplemented with graphite followed by exfoliation using a L5M high-shear laboratory mixer [29]. The prepared FFG was initially characterized by TEM, SEM, XRD, and Raman spectroscopy. TEM and SEM analyses were performed to examine the quality and exfoliation behavior of the prepared graphene sheets after exfoliation of graphite. Here, it can be clearly inferred from the TEM micrograph that the graphene nanosheets were packed together and well exfoliated, as shown in Figure 2. The smooth planar surface is free of defects and the edge is partially scrolled, which is due to the ultrathin graphene nanosheets. In addition, the thin, irregular curly shape in the SEM images confirms the exfoliation of graphite [17]. Further, SEM-EDX analysis confirms the presence of carbon and oxygen in the FFG sample, while an additional peak of Ag along with these two can clearly be observed for Ag–FFG (Figure 3).

Exfoliation was confirmed using a Raman spectrum. The 2D band at 2717 cm^−1^ depicts the layer number and thickness of the sample (Figure 4). In the present study, a significant and single 2D peak for FFG can be observed, which is more intense and sharper than that of expanded graphite (Figure 4b). This peak of FFG is due to the presence of two phonons with the opposite momentum at the highest optical branch near the K point of the Brillouin zone [17]. In addition, a slight shift in the 2D peak towards the left (as indicated in blue in Figure 4b) was also observed after the exfoliation process. The intensity ratio of D band to G band is commonly utilized to evaluate the degree of structural defects (I_D_/I_G_) in graphene flakes [17,41]. During an initial shear exfoliation study, the I_D_/I_G_ defect ratio was found to be 0.27, which is much lower than the defect ratio of 0.43 previously reported by Deshmukh and Kim [17] using the sonication method.

Figure 5 shows the XRD patterns of expanded graphite, FFG, and AgFFG. FFG has two diffraction peaks. There is a strong sharp diffraction peak located at 2θ = 26.5° (002) with an interlayer distance of 0.332 nm, which was determined using Bragg’s law equation. A weak diffraction peak located at 2θ = 54.6° (004) confirmed that the structure of graphene was well retained even after the exfoliation process, which corresponds to the highly organized crystal structure of hexagonal graphite [42]. Here, the peaks representing (020) and (004) correspond to graphene nanoplatelet flakes composed of a mixture of few- and multi-layered graphene [43]. The diffraction peaks at 38.1°, 44.2°, 64.3°, and 77.3° indicate (111), (200), (220), and (311) diffraction of metallic Ag, respectively. The AgFFG (with an interlayer distance of 0.338 nm) exhibited a peak related to FFG, and the peak position shifted slightly toward a lower 2θ value, confirming the production of AgFFG nanocomposites [44]. In addition, based on Scherrer’s equation, the crystal sizes of FFG and AgFFG were estimated to be 13.31 nm and 14.79 nm, respectively.

XPS was assessed to investigate metal percentages and bonding configuration. The XPS survey spectrum of AgFFG showed three characteristic peaks at C 1s, O 1s, and Ag 3d, (Figure 6a). Figure 6b shows the C 1s spectrum of AgFFG. The binding energies of the C 1s fitted peaks are 285.2, 286.4, 288.4, and 292.0 eV due to the sp^2^ carbon C-C, C-O, C=O, and O-C=O groups, respectively [17,45]. As shown in Figure 6c, the O 1s spectrum can be fitted to three peaks at 531.5, 532.5, and 533.7 eV due to the presence of various oxygen functionalities, such as C=O, C-O, and O=C-O, respectively [46]. As shown in Figure 6d, the Ag 3d region of AgFFG displayed two large typical peaks at binding energies of 386.4 and 374.9 eV, assigned to Ag 3d _5/2_ and Ag 3d _3/2_ [47]. One small peak located at 374.16 eV represents silver ions (Ag^+^) [48,49].

### 3.2. Optimization of Exfoliation Process

The effect of changing parameters on the yield of FFG was studied during the exfoliation method. Various parameters were optimized in the high-shear mixer: (i) rotational speed (3000–6000 rpm); (ii) exfoliation time (2–8 h); (iii) GNE concentration (2–10 g/L); and (iv) graphite concentration (5–30 g/L). 

#### 3.2.1. Effect of Rotational Speed

Initially, four different rotational speeds (3000, 4000, 4500, and 6000 rpm) in the presence of GNE were optimized for shear exfoliation. Figure 7 shows the Raman spectra at different rpm of shear exfoliation. As the rpm increased, the intensity and shape of the Raman spectrum of graphene changed significantly. It can be seen from Figure 7b that there is a direct effect of rpm on the left shift of the 2D band. As the rpm increased from 3000 rpm to 6000 rpm, the shift of the 2D peak gradually increased. Table 1 shows the quality of FFG derived from Raman spectroscopy at various rpm in GNE solution. It can be clearly observed that the number of FFG layers decreases as the rpm increases, i.e., from 8.610 to 5.248. At a maximum rpm of 6000, the lowest graphene layer of 5.248 was obtained. Due to the limitation of the rotational speed of the instrument, 6000 rpm was determined as the optimal rotational speed for further experiments. In addition, the degree of structural defects in the samples is indicated by the defect ratios (I_D_/I_G_), ranging from 0.220 to 0.391, and lateral flake sizes, ranging from 0.559 to 1.276 (Table 1).

#### 3.2.2. Effect of Exfoliation Time

The Raman spectra as a function of exfoliation time in the shear exfoliation process are shown in Figure 8. The shape and intensity of the Raman spectra of FFG were completely different from those of graphite. As the exfoliation time increased, the 2D band slowly shifted to the left, but the intensity of the peak remained the same. Table 2 shows the quality of FFG with exfoliation time. The exfoliation time was increased from 2 h to 8 h and the number of graphene layers was calculated by Raman analysis. As the exfoliation time increased from 2 h to 6 h, the number of FFG layers decreased from 9.750 to 5.236. At an exfoliation time of 6 h, the maximum exfoliation of graphite was observed. The structural defect degree of the sample showed a defect ratio (I_D_/I_G_) of 0.183 to 0.645, and the lateral flake size was 1.763 to 1.227. As the exfoliation time was further increased from 6 h, excessive damage to the quality and productivity of graphene occurred. Therefore, in this study, the optimal exfoliation time was determined to be 6 h.

#### 3.2.3. Effect of Initial Graphite Concentration and GNE Concentration 

The effect of GNE concentration and initial graphite concentration was investigated in the shear exfoliation process. Figure 9a shows the effect of the GNE concentration on graphite exfoliation. It is clear from Figure 9a that the FFG concentration increases significantly after 6 h of exfoliation as the GNE concentration increases from 2 g/L to 8 g/L. An FFG concentration of 1.369 g/L was achieved with a GNE concentration of 8 g/L, which is higher than previously reported study using the sonication method [50]. Figure 9b also shows the effect of the initial graphite concentration on the FFG concentration. Initially, graphite concentrations were studied in the range of 5–30 g/L. However, it was found that the exfoliation efficiency decreased as the initial graphite concentration increased to >10 g/L. The concentration of FFG was determined based on the Lambert–Beer law (i.e., A = α × C × l), where α, C, and l are the absorption coefficient, solute concentration, and light path length, respectively. The absorbance of FFG at 660 nm is not affected by the absorption of GNE because the absorption peak of GNE occurs at 310 nm. The value of the absorption coefficient α was determined to be 824.03 mg^−1^m^−1^mL. α depends on multiple factors, including lateral size, the number of monolayers per flake, the number and type of defects, and impurities in the graphene structure.

### 3.3. Antibacterial Activity of the Prepared Ag–FFG 

After the optimization of the exfoliation process, Ag–FFG was prepared as described in Section 2.3. Figure 3 clearly shows the presence of elemental Ag peaks on the surface of FFG along with carbon and oxygen, indicating the formation of Ag–FFG composites. For healthcare-associated infectious diseases, the possibility of selecting multi-drug-resistant microbes increases after treatment. The most frequent etiological agents are Gram-negative (*E. coli* and *P. aeruginosa*) and Gram-positive (*S. aureus* and *Bacillus* sp.) bacteria. Hence, these four microbes were selected in order to determine the behavior of synthesized Ag–FFG nanohybrids with regard to bacterial resistivity.

The antibacterial potential of Ag–FFG was evaluated against four different Gram-negative and Gram-positive bacterial species via the disc diffusion agar method (Figure 10). Regions of inhibition were clearly visible for Ag and Ag–FFG loading, whereas FFG showed no antibacterial properties. In the case of Ag–FFG, the Gram-negative bacteria *E. coli* and *P. aeruginosa* showed inhibitory zones of 0.048 mm/μg Ag and 0.19 mm/μg Ag, respectively, whereas for the Gram-positive bacteria *S. aureus* and *Bacillus* sp., the inhibitory zones were 0.238 and 0.048 mm/μg Ag, respectively. In the case of only Ag, the inhibitory zones were 0.064 and 0.032 mm/μg Ag for Gram-positive bacteria, *S. aureus* and *Bacillus* sp., respectively, while the Gram-negative bacteria, *E. coli* and *P. aeruginosa*, showed inhibitory zones of 0.047 mm/μg Ag. These results indicate that the antimicrobial activity of Ag–FFG depends on the bacterial type and is similar to or higher than that of Ag.

A plausible mechanism of Ag–FFG includes adhesion of Ag–FFG to microbial surfaces mediated via electrostatic interactions between negatively charged bacterial cell membranes and positively charged Ag–FFG [26,27]. Once Ag–FFG attaches to the bacterial surface, it penetrates the cells, resulting in the damage of cellular organelles. Attached Ag to the nanocomposites is also capable of generating reactive oxygen species and further can induce oxidative stress in the bacterial cells owing to the release of Ag^+^ ions. Ag attached to the FFG can interact well with sulfur-containing proteins (thiol groups of proteins, which are responsible for enzymatic activity) present on the bacterial membranes and inside them as well [27]. These chemical interactions can result in the inactivation of such proteins in the bacterial cell wall and are lethal.

Further, AgNPs possess a higher surface-to-volume ratio, which enhances their antibacterial activity. However, the aggregation of these nanoparticles greatly decreases their antibacterial activity. In the present work, Ag was loaded onto the FFG surface owing to the efficient prevention of Ag aggregation and the protection of high surface reactivity, whereas FFG has stronger adsorption properties due to its negatively charged capping molecules. Bacterial cells can be adsorbed onto the FFG surface to enhance the contact between the bacteria and the as-synthesized Ag possessing nanocomposites [24,26,27]. Therefore, Ag–FFG nanocomposites can inhibit bacterial growth very efficiently with improved antibacterial properties. This study shows that Ag–FFG can be used as an antibacterial nanomaterial. The FFG prepared in this study can be more advantageous for biomedical applications because it is produced in an environmentally friendly process without using toxic compounds, and the surface of graphene is functionalized with phytochemicals.

## 4. Conclusions

In the present investigation, we thoroughly demonstrated a simple, environmentally friendly and effective exfoliation method for converting graphite into functionalized few-layer graphene. FFG was successfully produced by a facile green one-step process. All selected parameters had a significant impact on the yield of FFG. XPS analysis highlighted the presence of a strong interaction between Ag and FFG, so that Ag was well dispersed onto the FFG surface. The maximum conversion through the high-shear exfoliation process occurred at an initial graphite concentration of 10 g/L with 8 g/L of gallnut extract at a rotational speed of 6000 rpm after 6 h. Using high-shear exfoliation, a FFG concentration of 1.36 g/L was obtained in a shorter time compared to previously reported sonication method. In addition, the prepared Ag–FFG showed antibacterial activity against both Gram-positive (*S. aureus* and *Bacillus* sp.) and Gram-negative bacteria (*E. coli* and *Pseudomonas aeruginosa*). Therefore, this study demonstrated a clear systematic overview of FFG production that can be further utilized in biomedical applications such as bioimaging, biosensors, dosimetry, drug delivery, tissue scaffolds, nanocarriers, etc.

## Figures and Tables

**Figure 1 micromachines-13-01232-f001:**
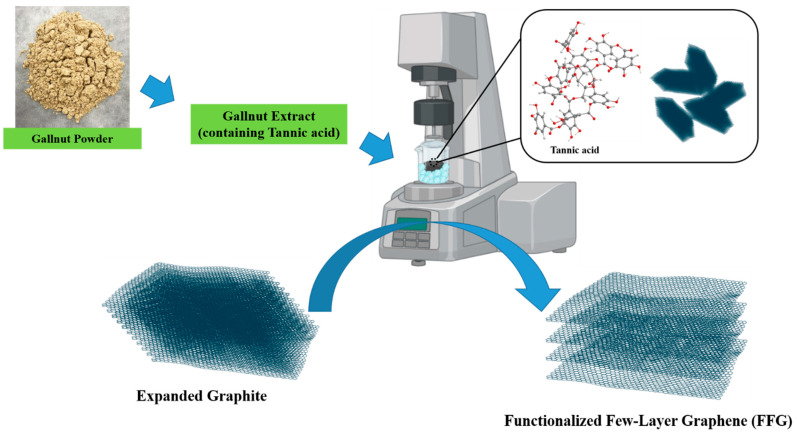
Schematic diagram for the preparation process of FFG.

**Figure 2 micromachines-13-01232-f002:**
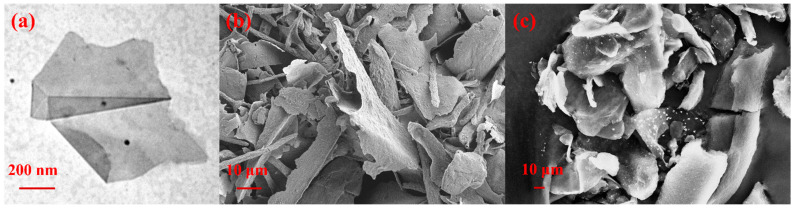
(**a**) TEM and (**b**) SEM micrographs of FFG; (**c**) SEM micrograph of Ag–FFG.

**Figure 3 micromachines-13-01232-f003:**
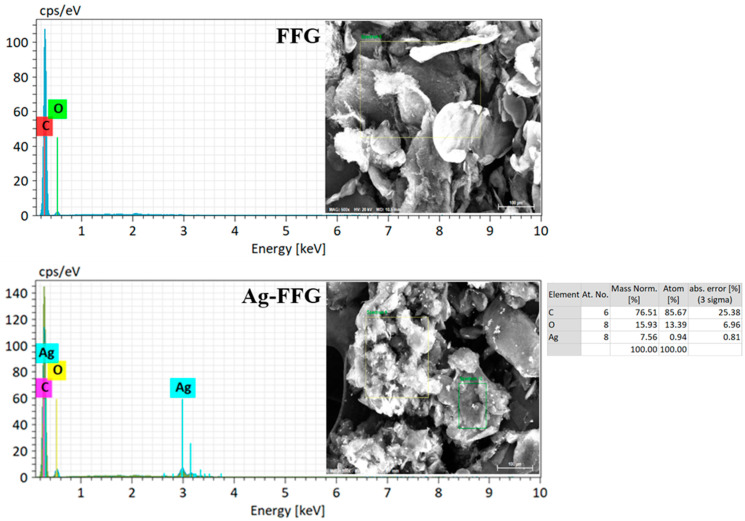
EDX spectra of FFG and Ag–FFG.

**Figure 4 micromachines-13-01232-f004:**
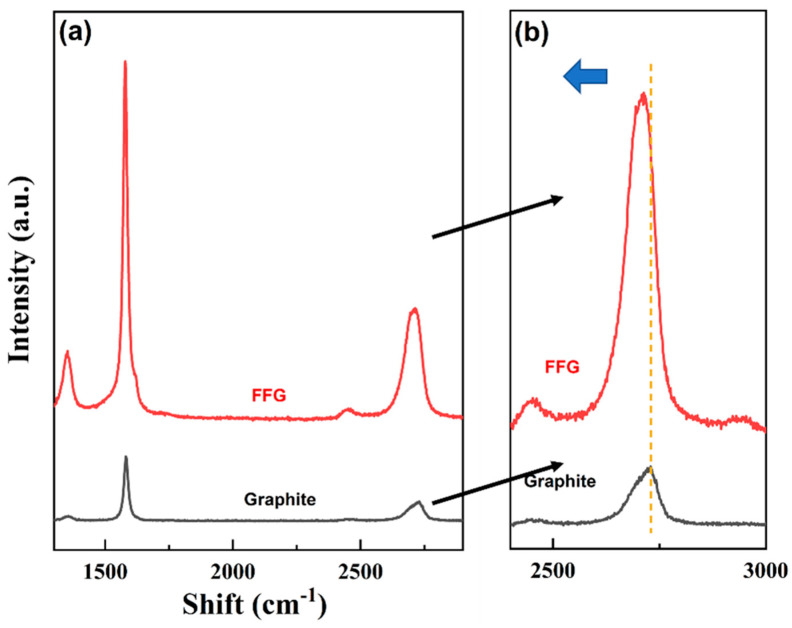
(**a**) Raman spectra and (**b**) magnified 2D bands of the spectra from (**a**).

**Figure 5 micromachines-13-01232-f005:**
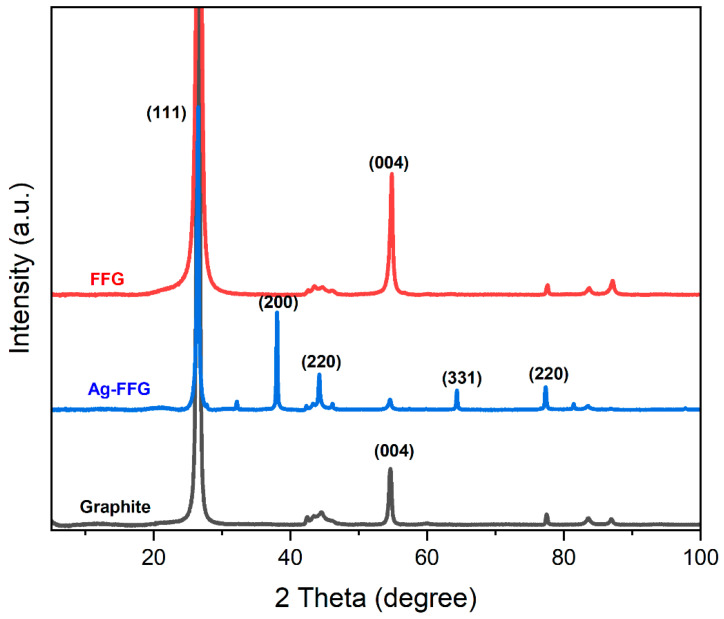
XRD patterns of graphite, FFG, and AgFFG.

**Figure 6 micromachines-13-01232-f006:**
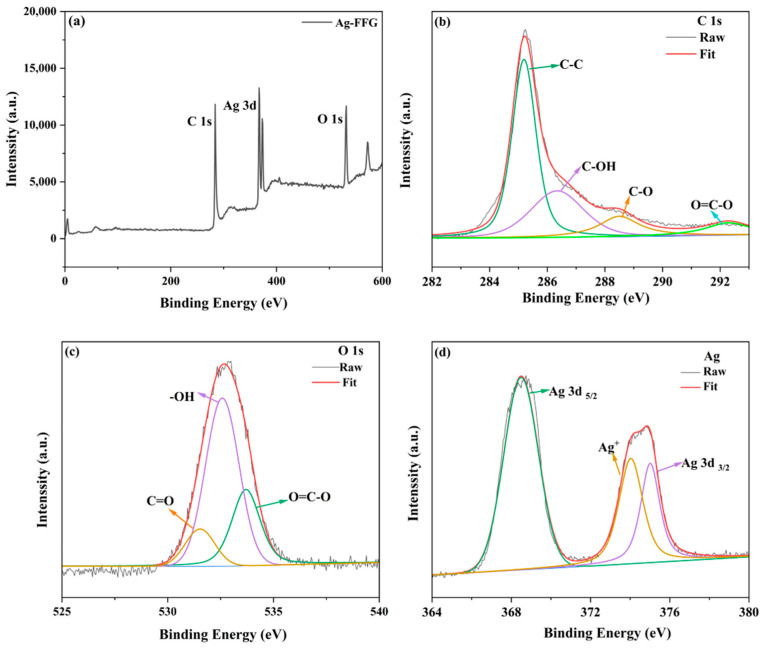
XPS characterization of AgFFG. (**a**) XPS survey spectra of AgFFG; (**b**) high-resolution C1s XPS spectrum; (**c**) high resolution O1s XPS spectrum of AgFFG; (**d**) high-resolution Ag3d XPS spectrum of AgFFG.

**Figure 7 micromachines-13-01232-f007:**
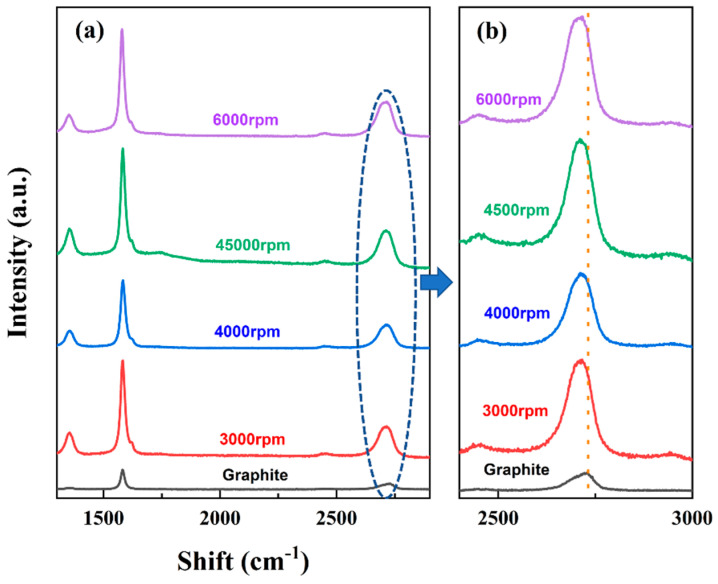
(**a**) Raman spectra of the prepared FFG at different rotational speeds using shear exfoliation; (**b**) magnified 2D bands of the spectra.

**Figure 8 micromachines-13-01232-f008:**
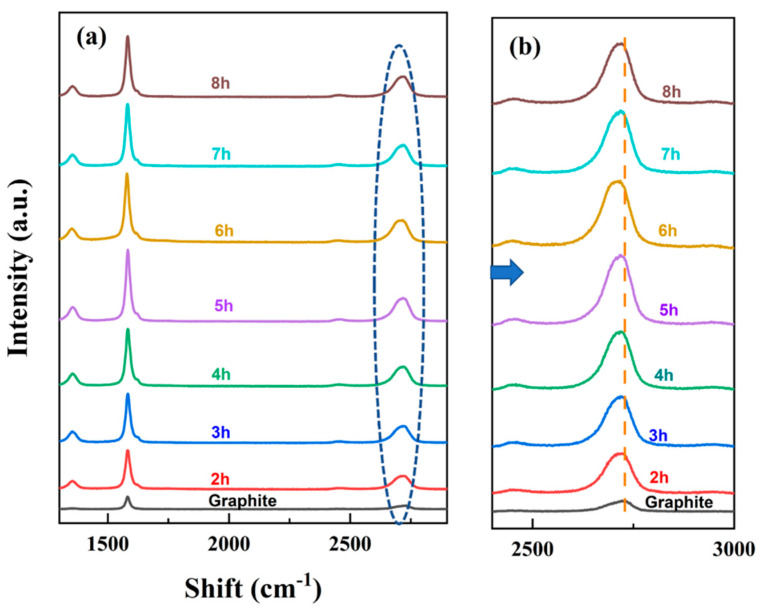
(**a**) Raman spectra of the prepared FFG at different exfoliation time; (**b**) magnified 2D bands of the spectra.

**Figure 9 micromachines-13-01232-f009:**
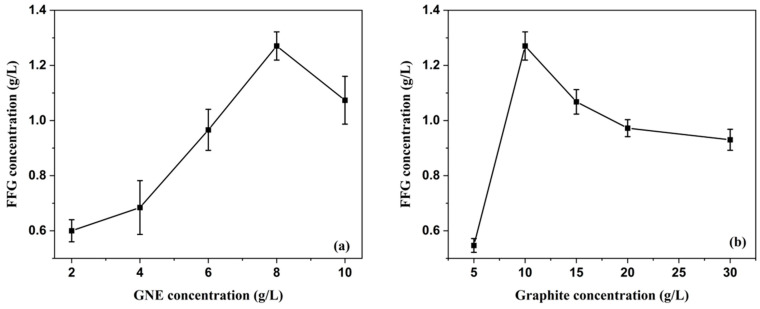
(**a**) Effect of GNE concentration and (**b**) initial graphite concentration on FFG concentration during the exfoliation process.

**Figure 10 micromachines-13-01232-f010:**
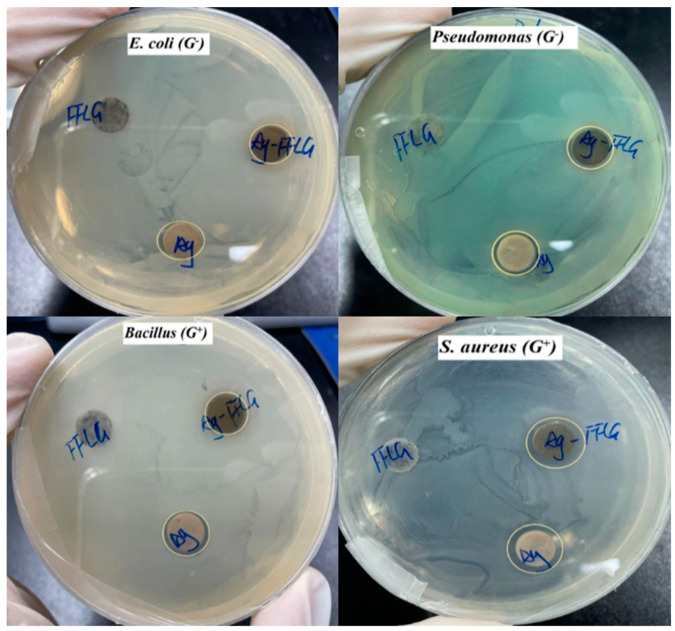
Antibacterial activity test of the prepared FFG, Ag–salt, and Ag–FFG against Gram-positive and Gram-negative bacteria.

**Table 1 micromachines-13-01232-t001:** Effect of rotational speed of high-shear mixer on FFG quality indicators determined using Raman spectroscopy.

RPM	*M* from Equation (1)	*N_G_* from Equation (2)	*L* (μm) from Equation (3)
3000	0.783	8.610	1.276
4000	0.753	7.727	0.936
4500	0.751	7.691	0.76
6000	0.638	5.248	0.559

**Table 2 micromachines-13-01232-t002:** Effect of exfoliation time on FFG quality indicators determined from Raman spectroscopy.

Exfoliation Time (h)	*M* from Equation (1)	*N_G_* from Equation (2)	*L* (μm) from Equation (3)
2	0.817	9.705	1.763
3	0.777	8.394	1.598
4	0.749	7.638	1.319
5	0.735	7.261	1.276
6	0.659	5.623	1.227
7	0.645	5.358	1.207
8	0.638	5.248	1.187

## Data Availability

The study did not report any data.

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
