# Peer review of "Green Production of Functionalized Few-Layer Graphene–Silver Nanocomposites Using Gallnut Extract for Antibacterial Application"

_micromachines, 2022, doi:10.3390/mi13081232_

Round 1
Reviewer 1 Report
The text needs more work

Author Response
|
S. No. |
Reviewer’s Comment |
Authors Response |
|
The authors investigated the process of obtaining graphene containing silver particles for antiseptic purposes. |
||
|
1. |
Fundamentally new in the study is the use of natural acid in the process of graphite exfoliation. Hence the question, if the same exfoliation conditions are used without this acid, will a graphene have the same characteristics? Graphene, like activated carbon, adsorbs on its surface everything that surrounds it, including microbes. Silver is a well-known antibacterial agent, especially colloidal silver, and is even used in toothpastes. From this point of view, there is nothing new. |
As rightly pointed by the Reviewer, the utilization of tannic acid from gallnut extract in the high shear exfoliation of graphite to FFG and Ag-FFG is the novelty of the present study. If the same exfoliation conditions will be used without tannic acid, there will definitely change in the characteristic properties of the prepared graphene.
Here, within the single process we are simultaneously performing the surface modification of graphite followed by the reduction to attain the functionalized graphene. Without tannic acid, the surface functionalization could not be possible. |
|
2. |
The introduction should indicate the primary sources in which the properties of graphene were first studied. And only the application is an achievement of recent years and there may be new, specific properties. |
As per the Reviewer suggestion, the primary sources in which the properties of graphene were initially studied along with their specific properties and applications are included in the revised manuscript.
Kindly see:
Page 1, Lines 30-33. |
|
3. |
There is not a word about silver in the introduction. Areas of its application. Including on the surface of carbon particles. |
The inclusion of the term “silver” along with the respective area of application on to the carbon surface has now been included in the revised manuscript. Kindly see:
Page 2, Lines 64-78. |
|
4. |
As for the figures, there should be numbers on the x-axis |
As rightly suggested by the Reviewer, the Authors have now included the scale of the X-axis for the figures in the revised manuscript. |
|
5. |
The scale must be transferred to the photo field. |
The improvement has been included in the revised manuscript. |
|
6. |
Superscript not in place |
Corrected, thank you. |
|
7. |
The conclusion needs to be specified and reflect the novelty of the study |
As per the Reviewer suggestion, the conclusion portion have been improved in the revised manuscript.
Kindly see:
Page 13, Lines 367-370. |
Reviewer 2 Report
In this manuscript, the author reported Ag loaded FFG, which obtained from expand graphite with GNE, under high-shear exfoliation. The prepared FFG was characterized using Raman spectra, XRD, TEM, and SEM, the Ag-FFG was confirmed by XRD and EDX, and the Ag-FFG composite exhibited antibacterial activity against both Gram-positive and Gram-negative bacteria. However, the whole manuscript was in bad organizing and writing, therefore a rejection of this manuscript is recommended.
(1) The Ag-FFG nanocomposites shown antibacterial activity, rather than raw FFG obtained with high-shear exfoliation. Therefore, the title of this work should be revised.
(2) Line 70-85, “2. Materials and Methods....” is writing guidelines in original template, which should be deleted.
(3) As a scientific report, the gallnut extract is present as tannic acid in Figure 1 is inappropriate. What is the main component of gallnut extract? And what is the content? These are very important for the repeating experiment. Therefore, FTIR or other characterization must be provided and analyzed.
(4) If the extraction procedure (100 °C for 90 min ) will damage the active component of gallnut? In addition, the references of extraction procedure should be provided.
(5) In line 124, What is the mechanism for the reduce of Ag+ to Ag? What is the reducing agent? Does the reaction occurred at room temperature?
(6) The scale ruler of TEM and SEM should be revised, which is too vague. In addition, the size of obtained FFG and Ag nanoparticles should provided.
(7) The antibacterial activity is highly dependent on the content of Ag, the content of Ag should be provided according to the EDX results.
(8) In the section of “3.2.3. Effect of initial graphite concentration and GNE concentration”, the author gave the result of the obtained graphene concentration, how does the author determination the graphene concentration? The determination method was not present in the section of “2.4. Characterization of FFG and Ag-FFG nanocomposites ”. In fact, the graphene concentration is hard to determination, as aromatic gallnut extract was adsorbed on the obtained graphene.
(9) The author provided the antibacterial activity of FFG, Ag-salt and Ag-FFG for four kinds of bacterial species, however the diameter of inhibition zone was not calculated. According to Figure 9, the diameter of inhibition zone for Ag-FFG is obviously smaller than that of pure Ag, thus the antibacterial activity of Ag-FFG is lower than that of pure Ag, the loading of Ag onto graphene is meaningless.
(10) The DOI number of the references should be provided.
Author Response
|
S. No. |
Reviewer’s Comment |
Authors Response |
|
In this manuscript, the author reported Ag loaded FFG, which obtained from expand graphite with GNE, under high-shear exfoliation. The prepared FFG was characterized using Raman spectra, XRD, TEM, and SEM, the Ag-FFG was confirmed by XRD and EDX, and the Ag-FFG composite exhibited antibacterial activity against both Gram-positive and Gram-negative bacteria. However, the whole manuscript was in bad organizing and writing, therefore a rejection of this manuscript is recommended. |
||
|
1. |
The Ag-FFG nanocomposites shown antibacterial activity, rather than raw FFG obtained with high-shear exfoliation. Therefore, the title of this work should be revised. |
As per the Reviewer suggestion, the title of the manuscript has been revised to:
“Green Production of Functionalized Few-Layer Graphene-Silver Nanocomposites Using Gallnut Extract for Antibacterial Application” |
|
2. |
Line 70-85, “2. Materials and Methods....” is writing guidelines in original template, which should be deleted. |
The writing guidelines was a typo-error and has been omitted from the revised manuscript. |
|
3. |
As a scientific report, the gallnut extract is present as tannic acid in Figure 1 is inappropriate. What is the main component of gallnut extract? And what is the content? These are very important for the repeating experiment. Therefore, FTIR or other characterization must be provided and analyzed. |
As rightly observed by the Reviewer, tannic acid is the chief component of the gallnut extract that has also been indicated in Figure 1. We have already performed a prominent study on to the characterization of the gallnut extract using the FTIR in our previous reported study, and has been referred at many places in the present manuscript.
“Deshmukh, A. R., & Kim, B. S. (2021). Bio-functionalized few-layer graphene for in situ growth of gold nanoparticles, improvement of polymer properties, and dye removal. Journal of Cleaner Production, 310, 127515.” |
|
4. |
If the extraction procedure (100 °C for 90 min) will damage the active component of gallnut? In addition, the references of extraction procedure should be provided. |
Gallnut extract mostly contains tannic acid. In addition, the melting point of tannic acid (280°C) is very high in comparison to the boiling point of water. Therefore, there will not be any de-nature of gallnut extract. Also, the respective reference has now been provided for the extraction process. Kindly see: Page 3, Line 98 |
|
5. |
In line 124, What is the mechanism for the reduce of Ag+ to Ag? What is the reducing agent? Does the reaction occurred at room temperature? |
The mechanism of reducing Ag+ to Ag has now been included in the revised manuscript. No, the reaction was performed at 60 ºC and has now been mentioned and highlighted in the revised manuscript. Kindly see: Page 4, Lines 126, 129-130 |
|
6. |
The scale ruler of TEM and SEM should be revised, which is too vague. In addition, the size of obtained FFG and Ag nanoparticles should provided. |
As rightly pointed out by the Reviewer, the scales for both the TEM and SEM micrographs along with the Ag-FFG has been provided in the revised manuscript.
Kindly see Figure 2 in the revised manuscript. |
|
7. |
The antibacterial activity is highly dependent on the content of Ag, the content of Ag should be provided according to the EDX results. |
As per the Reviewer suggestion, the content of Ag using the EDX results has now provided in the revised manuscript and Figure 3. |
|
8. |
In the section of “3.2.3. Effect of initial graphite concentration and GNE concentration”, the author gave the result of the obtained graphene concentration, how does the author determination the graphene concentration? The determination method was not present in the section of “2.4. Characterization of FFG and Ag-FFG nanocomposites”. In fact, the graphene concentration is hard to determination, as aromatic gallnut extract was adsorbed on the obtained graphene. |
Authors thanks the question raised by the Reviewer. Here, “the concentration of FFG was determined based on the Lambert-Beer law (i.e., A = α× C × l); where, α, C and l are the absorption coefficient, solute concentration and light path length, respectively. The absorbance of FFG at 660 nm is not affected by the absorption of GNE because the absorption peak of GNE occurs at 310 nm. The value of the absorption coefficient α was 824.03 mg−1 m−1 mL.
This explanation is also been added in the revised manuscript.
(Kindly see: Page 11, Lines 308-314)
Also, we truly respect the Reviewer comments that determination of the graphene concentration is very hard, as the aromatic gallnut extract was adsorbed on the obtained graphene. But we have very clearly mentioned in the methodology section:
(Kindly see: Page 4, Lines 121-123)
“To remove unbound extracts or free phytochemicals, the FFG nanosheets were washed repeatedly with distilled water, centrifuged at 13000 rpm for 20 min, and finally the precipitated sheets were freeze dried to obtain FFG powder.” |
|
9. |
The author provided the antibacterial activity of FFG, Ag-salt and Ag-FFG for four kinds of bacterial species, however the diameter of inhibition zone was not calculated. According to Figure 9, the diameter of inhibition zone for Ag-FFG is obviously smaller than that of pure Ag, thus the antibacterial activity of Ag-FFG is lower than that of pure Ag, the loading of Ag onto graphene is meaningless. |
The AgNPs possess a higher surface-to-volume ratio that enhances the antibacterial activities. However, the aggregation of these nanoparticles greatly decreases their antibacterial activities. In the present work, the Ag have been loaded onto the FFG surface owing to the efficient prevention of Ag aggregation and protect their high surface reactivity whereas FFG has the stronger adsorption properties due to the negative charged capping polymers, and the bacterial cells could be adsorbed onto the FFG surfaces that enhances the contact between the bacteria and the as-synthesized Ag possessing nanocomposites.
Also, as per the EDX analysis, it is very clear now that the Ag-FFG composite contains 7.56% of the Ag content. That means with low Ag content we can attain the anti-bacterial zone. Therefore, the loading of Ag on to the FFG is totally worthful. The diameter of inhibition zones per μg of Ag were calculated for Ag-FFG and Ag only. The results indicated that the antimicrobial activity of Ag-FFG depends on the bacterial type, and is similar to or higher than that of Ag.
|
|
10. |
The DOI number of the references should be provided. |
All the DOI for the references is now included in the revised manuscript. |
Reviewer 3 Report
The concept of the manuscript is to prepare functionalized few-layer graphene, through high-shear exfoliation using gallnut extract. The preparation of functionalized few-layer graphene using gallnut is not novel. However, linking FFG with Ag NPs is an interesting concept. I recommend the manuscript for publication after major revision.
1) In Figure 4 the scale of the axe X should be added.
2) The authors have to precisely determine the interlayer distance of the produced FFG and Ag-FFG and the crystallite's size, as well. The simplest way is through the Bragg's law, while the crystallite's size through the Scherrer equation.
3) XPS should be provided to confirm the bonding type between Ag and FFG.
4) The authors have to explain and discuss Figure 9 in a plainer way.
Author Response
|
S. No. |
Reviewer’s Comment |
Authors Response |
|
The concept of the manuscript is to prepare functionalized few-layer graphene, through high-shear exfoliation using gallnut extract. The preparation of functionalized few-layer graphene using gallnut is not novel. However, linking FFG with Ag NPs is an interesting concept. I recommend the manuscript for publication after major revision. |
||
|
1. |
In Figure 4 the scale of the axe X should be added. |
As rightly suggested by the Reviewer, the Authors have now included the scale of the X-axis for Figure 4 in the revised manuscript. |
|
2. |
The authors have to precisely determine the interlayer distance of the produced FFG and Ag-FFG and the crystallite's size, as well. The simplest way is through the Bragg's law, while the crystallite's size through the Scherrer equation. |
As suggested by the Reviewer, we have determined the interlayer distance of the produced FFG and Ag-FFG along with the crystallite size in accordance with the Bragg’s Law and Scherrer equation in the revised manuscript.
Kindly see:
Page 5, Lines 170-182;
Page 7, 230-242. |
|
3. |
XPS should be provided to confirm the bonding type between Ag and FFG. |
As per the Reviewer suggestion, we have performed the XPS analysis for determining the bonding type between Ag and FFG and the methodology and results and discussion have been included in the revised manuscript.
Kindly see:
Page 5, Lines 180-182; Page 7, 243-252 |
|
4. |
The authors have to explain and discuss Figure 9 in a plainer way. |
As per the Reviewer suggestion, we have tried to re-form the sentences explaining the Figure 9 and elsewhere required throughout the manuscript in the plainer way.
Kindly see:
Page 11-12, Lines 329-357. |
Round 2
Reviewer 2 Report
The paper was corrected again taking into account all my comments. Now the paper is much better compaed to the first version and in this form can be published in the presented form.
Reviewer 3 Report
The authors have addressed my comments and thus I recommend the manuscript for publication in its current form.